# A Review on the Effect of the Mechanism of Organic Polymers on Pellet Properties for Iron Ore Beneficiation

**DOI:** 10.3390/polym14224874

**Published:** 2022-11-12

**Authors:** Hongxing Zhao, Fengshan Zhou, Hongyang Zhao, Cunfa Ma, Yi Zhou

**Affiliations:** 1Beijing Key Laboratory of Materials Utilization of Nonmetallic Minerals and Solid Wastes, National Laboratory of Mineral Materials, School of Materials Science and Technology, China University of Geosciences (Beijing), No. 29 Xueyuan Road, Haidian District, Beijing 100083, China; 2Key Laboratory of Biomedical Polymers, Ministry of Education, College of Chemistry and Molecular Sciences, Wuhan University, Wuhan 430072, China; 3School of Materials Science and Engineering, Taiyuan University of Science and Technology, Taiyuan 030024, China

**Keywords:** organic polymers, bonding mechanism, pellet strength at low temperature, pellet bursting temperature, pellet strength at high temperature

## Abstract

Iron ore pellets not only have excellent metallurgical and mechanical properties but are also essential raw materials for improving iron and steel smelting in the context of the increasing global depletion of high-grade iron ore resources. Organic polymers, as important additive components for the production of high-quality pellets, have a significant impact on the formation as well as the properties of pellets. In this review, the mechanisms of organic polymers on the pelletizing properties, bursting temperature, and pellet strength at low and high temperatures, as well as the existing measures and mechanisms to improve the high-temperature strength of the organic binder pellets are systematically summarized. Compared with traditional bentonite additives, the organic polymers greatly improve the pelletizing rate and pellet strength at low temperatures, and significantly reduces metallurgical pollution. However, organic binders often lead to a decrease in pellet bursting temperature and pellet strength at high temperatures, which can be significantly improved by compounding with a small amount of low-cost inorganic minerals, such as bentonite, boron-containing compounds, sodium salts, and copper slag. At the same time, some industrial solid wastes can be rationally used to reduce the cost of pellet binders.

## 1. Introduction

Iron ore is an important raw material for steel production. According to statistics, world iron ore production increased from 1.043 Bt in 2001 to 2.93 Bt in 2012 [1]. Due to the increasing consumption of high-grade iron ore, the iron content in iron ore is decreasing, and these iron ores have to be finely ground and beneficiated to enrich the formation of iron concentrates, which are then further aggregated into pellets by binders to form pellet ores [2]. Pellets have the advantages of uniform particle size, high strength, suitability for long-distance transportation and storage, high iron grade, good metallurgical properties, and are conducive to improving the permeability of the material column and reducing the coke ratio during smelting [3]. In European countries, the proportion of pellet ore in the blast furnace charge is as high between 80% and 90%, even up to 100%, while the average proportion in China is only about 15%. According to the Commodity Research Unit (CRU) regarding the global consumption of iron ore resources in 2017, the consumption of pellet ore was 443 million tons, accounting for approximately 22% of the total global iron ore consumption. In 2020, the production of pellet ore by the key steel enterprises of the China Iron and Steel Industry Association reached 108 million tons. With increasing steel production, the annual global consumption of pellet binder is very significant.

Pelletizing binders mainly include inorganic binders, organic binders, and composite binders [4], which not only bond iron concentrate powder into pellets in the pelletizing process but also significantly improve the pelletizing rate and the quality of green and dry pellets, playing a fundamental role in the metal smelting process [5]. At present, bentonite is still the main pellet binder in China, with an average addition of up to 2~3wt.%, much higher than the foreign level of 0.5~0.7wt.% [6]. According to production experience, for every 1% bentonite added, the iron grade of the pellet will decrease by approximately 0.6wt.% [7], and the iron content will decrease by approximately 7 kg/ton [8]. An increase of 1% of SiO_2_ content in the pellet will lead to an increase of 4~7 USD/ton in steelmaking cost [9], while the organic binder is mainly polysaccharide materials and does not contain inorganic components, such as SiO_2_ and CaO. In the high-temperature sintering stage of pellets, it is volatilized by thermal decomposition, and there is almost no residual silicate slag phase inside the pellet, which greatly reduces metallurgical pollution. The organic polymer is gradually becoming a new type of pellet binder that can completely or partially replace bentonite.

After a long period of continuous development, the types of organic pellet binder have been enriched, resulting in organic substances, such as sodium carboxymethyl cellulose (Na-CMC), modified starch, modified humic acid, polyacrylamide, ligninsulfonate, guar gum, and molasses [10,11]. These organic substances are more water-soluble and cementable, which significantly improves the sphericity and low-temperature strength of pellets. However, adding too much organic binder not only leads to a lower pellet bursting temperature but also decomposition by burning after high-temperature roasting, which leads to more residual pores and makes the pellet strength at high-temperature decrease [12]. Many pellet researchers obtained good results of fired pellet strength by adding inorganic minerals, such as bentonite, borate, and sodium salt [2], and industrial solid waste, such as copper slag [13,14]. Although some reviews have addressed the relevant contents, there is no systematic theoretical discussion on the mechanism of organic binders on the pelletizing properties, bursting temperature, and pellet strength at low and high temperatures, especially the lack of a summary of the mechanism of inorganic minerals to improve the pellet strength at high temperatures [15]. This review summarizes in detail the internal behavior of organic polymers in pellets and the high-temperature reaction properties of inorganic minerals at high and low temperatures, thus, providing a theoretical basis for further guidance on the synthesis of cost-effective organic pellet binders and the preparation of high-quality pellet ores.

## 2. The Process of Producing Iron Ore Pellet

The industrial pelletizing process consists of three steps: raw material preparation, green pellet formation and the induration process [16]. The raw material for pellet ore is mainly derived from magnetite or hematite, and these minerals need to be dried and ground by a high-pressure roller before they can be used for the preparation of green pellets. Grinding treatment can improve the physical properties of minerals such as particle size and specific surface area, which is beneficial to improve the capillary force between minerals and green pellet quality [17,18]. Due to the decreasing resources of high-quality iron ore, some iron concentrate often has to undergo flotation and magnetic separation to make the iron content greater than 65%. Pelletizing binder is an essential addition to the pelletizing process and can significantly improve the strength and metallurgical properties of pellets. After pretreatment, iron concentrate and binder are operated by mixer and pelletizer to form green pellets, which are then processed at a high temperature by a shaft furnace, chain grate machine—a rotary kiln or belt roaster—to form hardened pellets 12~15 mm in diameter. The process of producing iron ore pellet in factories of BAOWU is shown in Figure 1 [19].

## 3. The Binding Mechanism of Organic Binders in the Pellets

Green pellets are the most critical factor in producing high-quality pellet ore. The formation process usually includes three stages: cue pellet formation, cue pellet growth, and green pellet compaction, as shown in Figure 2 [20]. In the pelletizing process, the iron ore is moistened with water and the excess free energy on its surface attracts water molecules to form a layer of adsorbed water that cannot move freely. Under the rolling action of the disc pelletizer, the iron ore particles are compacted to form capillary water, which generates a capillary force [21,22], and the surrounding iron ore particles are pulled toward the center of the water droplet by the capillary force, thus, forming a tight particle bond, which is the cue pellet [23]. After the cue pellet is continuously rolled and pressed, the interior is gradually compacted and the capillary water is squeezed onto the surface of the cue pellet, further bonding the surrounding iron ore particles, resulting in continuous growth [24,25]. The growth of the cue pellet depends on the capillary effect, and its pelletizing rate depends on the migration rate of capillary water. When the cue pellet grows to a proper size, the water addition should be stopped and the excess water in the green pellet should be squeezed out by increasing the rolling, which is the green pellet compaction process [26].

The main components in the pellets are a lot of fine iron ore, a small amount of binder, and water. In the pelletizing process, there are complex interactions between iron ore particles, binders, and water molecules. In the iron-bentonite system, the capillary force is the dominant force in the formation of green pellets, and its size depends on the particle size distribution and hydrophilicity of iron ore [27,28]. Therefore, pellet materials, bentonite dosage, ball mill parameters, moisture, time, and operating conditions of the pelletizer all affect the formation and performance of the green pellets [29,30,31]. In the iron ore-organic system, cohesion and adhesion are the main forces that exist inside the pellet [32,33], capillary forces are almost negligible compared to them, and the internal forces acting inside the pellet are equivalent to the sum of cohesion and adhesion of the organic binder [34].

Cohesion is the net interaction force between binder molecules [35], and its magnitude depends on the chemical structure, molecular weight, crystallinity, and degree of cross-linking and branching of the organic material [36]. For organic binders to have strong cohesion, Qiu, Jiang, et al. [37] suggested that the polymer molecules must contain a strong organic chain backbone structure and a high degree of polymerization. By introducing double and triple bonds in the organic chain skeleton, adding aromatic rings, bulky substituents, and non-rotating groups in the main chain can make the molecular structure of organic compounds stronger. The higher degree of polymerization of organics increases the viscosity of the organic binder solution, which effectively controls the water transport inside the pellet and improves the pellet formation ability in an effective time [38]. Low viscosity leads to faster moisture agglomeration on the pellet surface, which may result in the uncontrolled and rapid growth of green pellets, forming small pellets with a rough surface and low strength [39]. High viscosity tends to slow down the water agglomeration and slow down the growth rate, allowing the pellets to be adequately compacted. Therefore, the ideal viscosity allows the cue pellets to coalesce with iron ore concentrate powder within an effective collision time [40]. However, cations such as Ca^2+^ and Mg^2+^ in water can shield the ionic repulsion of carboxyl groups in water-soluble anionic organic polymers, inhibiting their decomposition and extension in solution and reducing the viscosity of organic solutions [41,42]; the negative effect of cations can be reduced by adding appropriate amounts of Na_2_CO_3_ to precipitate Ca^2+^ in the form of CaCO_3_ [43].

Adhesion force refers to the chemical bond, hydrogen bond, electrostatic force, and magnetic force generated between organic binder and iron ore at the interface. The iron ore surface is hydroxylated upon contact with moisture and adsorbs H^+^ or OH^-^ from aqueous solutions, resulting in a surface with different electrical properties [44]. In acidic solutions, the surface hydroxylation of iron ore adsorbs H^+^ from aqueous solutions to form Fe-OH^2+^, making the surface positively charged. Some of the carboxyl functional groups in the organic polymer HO- [P]-COOH molecule ionizes to form HO- [P]-COO^-^, which exhibits negative electrical properties [45]. At this point, the organic polymer molecules and the mineral surface carry opposite charges, generating electrostatic attraction forces, while the two will further interact to form the HO-P-OC-O-Fe ligand exchange [46]. In alkaline solutions, the surface hydroxyl group of iron ore adsorbs OH^-^ from aqueous solutions to form Fe-O^-^, and the organic binder molecule HO- [P]-COOH carboxyl functional group combines with OH- to form HO- [P]-COO^-^, which exhibits a strong negative charge, and the organic binder molecule has the same charge as the mineral surface, generating electrostatic repulsion [47,48] (the internal force of the pellet is shown in Figure 3. Thus, the interaction forces, such as electrostatic attraction force, ligand exchange, and hydrogen bonding, greatly enhance the adhesion of the binder to the mineral particle surface.

However, it is far from enough for the organic binder to produce strong adhesion to the surface of iron ore; its solution should be able to quickly diffuse to the surface of iron ore when contacting minerals and have good wettability [49]. The diffusion coefficient ($S_{H}$) is an important index to measure the wettability of the solution. Qiu, Jiang, et al. [37] define it as the difference between adhesion work (WA) and cohesion work (WC), as shown below:$$S_{H}=W_{A}-W_{C}$$

This coefficient can also be expressed as $S_{H}=\gamma_{lv}\left(\cos\theta -1\right)$.

In the formula, $\gamma_{lv}$ is the liquid/gas interfacial tension and θ is the contact angle. It can be seen from the equation that when $\gamma_{lv}$ is kept constant, the smaller the contact angle, the larger the diffusion coefficient, so the θ between the organic binder solution and the iron ore surface should be minimized. The contact angle depends on the characteristics of the binder solution and the iron ore surface. Introducing hydrophilic and solid functional groups in the molecular structure of the organic binder, reducing the viscosity of the binder solution, and increasing the density of hydroxyl groups on the iron ore surface will reduce the contact angle and improve the wettability of the iron ore surface with the organic binder solution [50].

It can be seen that the organic binder absorbs water and forms a solution with certain viscosity, which not only wraps or fills in between the iron ore particles but also improves the interaction force between the polymer and iron ore by forming hydrogen bonds, electrostatic attraction and ligand exchange with the mineral surface through functional groups, such as hydroxyl and carboxyl groups. At the same time, the organic binder improves the diffusion of the aqueous solution on the surface of the iron ore so that the organic binder can fully adhere to the iron ore in an effective time and improve the pelletizing rate. Therefore, the adhesion, cohesion, and diffusion of the organic binder are the important forces affecting the performance of pellet ore, and the preparation of the ideal organic binder must have the appropriate degree of polymerization as well as the type and number of polar and hydrophilic groups.

## 4. Effect of the Mechanism of Organic Polymers on the Pellet Strength at Low Temperature

The pellet strength at low temperatures includes green pellet drop number and dry pellet compressive strength. After the pelletizing process, the iron ore powder forms green pellets of uniform size. To facilitate transportation and processing, green and dry pellets must be of a certain strength. The important indicator used to evaluate the strength of green pellets is the green pellet drop number, which is generally required to be from 4–6 times/pellet, and the dry pellet strength is generally required to be ≥25 N/pellet. The strength of green and dry pellets is closely related to the adhesion and cohesion of organic binders, and its strength usually increases with the increase in organic binders, as shown in Figure 4 and Figure 5.

As can be seen from the figures, in the range from 0~2wt.% addition of organic polymers, the approximate order of the influence on the green and dry pellet strength is Floform 1049 V > Guar Gum > HV-CMC > LV-CMC > MHA > Na-LS > Corn starch > Dextrin. Floform 1049 V, Guar Gum, and HV-CMC have large molecular weight and high solution viscosity, which can easily form strong cohesion inside the pellet, while their molecular structures contain a large number of active functional groups such as amide groups, hydroxyl groups, and carboxyl groups, which can form strong adhesion with mineral surfaces. However, the cohesion of organic binders tends to contribute more to the strength of green and dry pellets than adhesion because organic polymers with large molecular weights can generate greater chain interactions between them, and these interactions increase with the degree of polymerization and branching, with longer and more branching chains becoming entangled due to the mutual entanglement of neighboring binder molecules allowing more stress dissipation [61]. However, there are also large differences in the magnitude of the forces adsorbed with the iron ore surface for organic substances containing different functional groups.

Floform 1049 V is a polyacrylamide (PAM)-based polymer produced by SNF. The PAM forms ligand bonds with Fe^3+^ on the surface of iron minerals mainly through lone pair electrons of nitrogen in the amide group and hydrogen bonds with hydrogen atoms of Fe-OH [62]. Hydrogen bonding is the main driving force controlling PAM adsorption, which occurs between the electronegative C=O group on the PAM and the donor proton oxide surface hydroxyl group, positive M-OH^2+^ and neutral M-OH [63,64]. Guar gum is a natural polymeric hydrocolloid with high viscosity, good water solubility, and a molecular structure containing mainly benzene rings and hydroxyl groups [65]. Guar gum readily forms a viscous colloid at lower doses and adsorbs on the surface of iron ore through hydroxyl groups in a bulk structure, forming an adsorbed layer that produces high green and dry pellet strength [58].

Carboxymethyl cellulose (CMC) is a water-soluble cellulose derivative produced by partial substitution of the hydroxyl group of cellulose by carboxymethyl [66]. CMC generates electrostatic force, van der Waals force, and hydrogen bonding adsorption with mineral particles through functional groups such as carboxyl and hydroxyl groups [67]. In higher moisture, the CMC of high molecular weight can extend sufficiently into the liquid to achieve the maximum thickening effect, and a higher degree of substitution increases solubility and reduces the interaction effect of the binder with various ions in the solution [68,69]. After drying, CMC can be adsorbed on the surface through the formation of chemical bonds between oxygen (-OH, -CH_2_COOH, or -CH_2_OCH_2_- backbone) present in the CMC molecules and Fe^3+^/Fe^2+^ on the iron ore surface [70].

Starch forms adsorption with the mineral surface mainly through hydrogen bonding, and the solubility of starch increases after artificial modification treatment [71], with the increase in starch solubility, water absorption capacity increases, starch dispersion improves, and solution viscosity becomes higher. Compared to the pellets without binders, the gelatinized starch particles adhere to the iron ore surface in large quantities and connect the iron ore particles through the starch gel matrix, thus, enhancing the binding effect of starch to iron ore concentrate powder, and the green and dry pellet strength increases [72], as shown in Figure 6a–c.

Modified humic acids pellet binder (MHA) contains a large number of functional groups such as carboxyl and hydroxyl groups, which are easily adsorbed on the surface of mineral particles, wrapping iron ore particles, and relying on the cohesion and adhesion generated by the humic acids network to enhance the bridging action between iron ore particles and tightly agglomerate them [75]. With the increase in humic acids (HA), the colloid effect is enhanced, causing the adsorption of iron ore particles such as ligand exchange and electrostatic interaction, which substantially increases the green pellet strength [76,77]. Numerous studies have shown that the adsorption of HA on magnetite/hematite is directly related to HA concentration, pH, and metal cations [78,79]. The adsorption density increases with increasing HA concentration and decreases with increasing pH [80]. Metal cations tend to cause Fulvic acids (FA) molecules to coalesce together, changing the stretching and curling state of FA itself and increasing the viscosity of FA solutions [81], resulting in higher green and dry pellet strength [82].

Lignosulfonate contains a large number of functional groups such as hydroxyl, sulfonic acid, and carboxyl groups [83,84], and the unshared electron pairs of oxygen atoms on their functional groups can form ligand bonds with Fe^3+^. Lignosulfonate solutions are prone to the formation of anionic groups with strong electronegativity, and the electrostatic repulsion between anionic groups will keep the solution in a highly stable dispersion. The main driving forces for ligninsulfonate adsorption at the solid–liquid interface are electrostatic force, hydrogen bonding, and van der Waals forces [85]. The network and benzene ring-like organic chains in the substructure of lignosulfonate have bonding properties, and a large number of carboxyl and sulfonic groups can have strong electrostatic interaction with the surface of iron concentrate, thus, improving the adhesion between iron ore and binder [86].

## 5. Effect of the Mechanism of Organic Polymers on Pellet Bursting Performance

The bursting temperature is an essential index for evaluating the quality of pellets. Usually, the temperature at which 1.0~2.0 m/s air velocity and 10% of the green pellets produce cracks or bursts is used as the bursting temperature of the pellets; in particular, the bursting temperature must be higher than 650 °C in the drying process of the shaft furnace. Increasing the bursting temperature can reduce the bursting of green pellets in the drying process, improve the permeability of the charge, and reduce caking accidents. Additionally, a higher bursting temperature can improve drying temperature and, thus, shorten drying time.

The main reason for bursting is the unbalanced effect of surface vaporization and the internal diffusion of green pellets. In the drying process, the surface of the pellets is first heated and vaporized, resulting in a moisture difference between the surface and the inside of the pellet, with a small surface humidity resulting in a large shrinkage and a large central humidity resulting in a small shrinkage. Such uneven shrinkage easily leads to the stress on the surface of the green pellets exceeding the ultimate strength, resulting in cracks [87]. When the surface of green pellets gets dried, the vaporized layer gradually spreads to the interior, at which time the moisture diffuses outside the surface through the capillaries of the drying layer. When the drying temperature is too high, the vaporization rate is too fast, and the vapor pressure inside the pellet increases because it cannot diffuse to the surface of the pellet in time, and when the vapor pressure exceeds the ultimate strength of the drying layer, the pellet will burst [88], as shown in Figure 7. Therefore, the main reason for bursting is that the steam inside the pellet does not diffuse to the surface effectively and timely, resulting in excessive internal vapor pressure.

Organic polymers, as widely used additive components in pellets, have a significant effect on pellet bursting temperature, as shown in Figure 8. It can be seen that almost all organic polymers will make the pellet bursting temperature show a changing trend of increasing first and then decreasing. The bonding strength of the binders, the viscosity of the binder solution, and the porosity of the dry pellet have direct effects on the bursting temperature [89]. Bentonite can significantly increase the bursting temperature because of its strong dry pellet strength and large porosity [90]. Compared to bentonite, the organic binder has a higher viscosity at room temperature, which can not only produce high dry pellet strength, but can also effectively reduce the diffusion rate of free and capillary water and the surface vaporization rate and slow down the degree of shrinkage unevenness caused by humidity differences, while volatilization of the organic polymer at high temperature will increase the porosity of pellets and promote the rapid diffusion of vapor, thus, reducing the vapor pressure inside the pellet and increasing the pellet bursting temperature [91]. However, the addition of a large amount of organic polymer will not allow the water vapor inside the pellet to be discharged from the pellet in time, increasing the vapor pressure inside the green pellet and, thus, a decrease in the bursting temperature [92]. Therefore, it is necessary to increase the outward diffusion rate of water vapor inside the pellets and maintain the balance between the vaporization on the surface of the pellets and the internal diffusion to increase the bursting temperature [93]. At the same time, the pellet bursting temperature can be increased by selecting suitable raw materials and binders, optimizing pelletizing parameters, and adding an explosion-proof agent [94,95].

## 6. Effect of the Mechanism of Organic Polymers on Pellet Strength at High Temperature

Under high-temperature conditions, pellet ore mainly undergoes two stages of preheating and roasting, while the strength of fired pellets is closely related to the recrystallization process of Fe_2_O_3_ and the consolidation of the liquid phase. The strength of the fired pellet is mainly affected by the characteristics of iron ore concentrate, the roasting process, additive compositions, and other factors. The general standard of the fired pellet strength is ≥2000 N/pellet, and the standard of the extremely large blast furnace must be above 2500 N/pellet.

The organic binder affected the fired pellet strength mainly by changing the porosity of the pellets, as shown in Figure 9. For PAM, CMC, Guar gum, and Na-LS, the fired pellet strength showed a tendency to increase and then decrease with the addition of binders, and an appropriate amount of organic polymer can improve the porosity of pellets, thus, promoting the solid-phase consolidation formed by the oxidative recrystallization of Fe_3_O_4_ [100]. Both corn starch and dextrin are starch products, and adding a small amount of these substances will cause iron oxide particles to be encapsulated in a continuous matrix or gel, and this organic matrix or gel structure is broken down to form loose particles at high temperatures compared to pellets without any added binders, leading to a decrease in the strength of fired pellets. Thereafter, as the number of starch products added increased, the trend of pellet strength changed similarly to that of PAM and CMC. MHA is distinguished from other organic polymers by the presence of trace amounts of inorganic mineral components that can induce the generation of a certain amount of high-temperature liquid phase, which compensates for the significant reduction in fired pellet strength due to increased porosity [101].

The oxidation and recrystallization of Fe_3_O_4_ is an important process to improve the fired pellet strength. In a fully oxidized atmosphere, adding an appropriate amount of organic matter can produce a certain number of pores inside the pellets, which is conducive to the full diffusion of O_2_ into magnetite pellets to oxidize Fe_3_O_4_ into the primary crystal of Fe_2_O_3_, as shown in Figure 10a. At the same time, as the temperature rises, the oxidized Fe^3+^ on the surface of the pellet will continue to promote the diffusion of Fe_2_O_3_ to the inner layer, and the primary grains of Fe_2_O_3_ grow up and get close to each other, and some of the grains will start to connect, gradually forming a Fe_2_O_3_ microcrystal connection bridge, as shown in Figure 10b. When the temperature reaches approximately 1250 °C, Fe_2_O_3_ crystals grow up again, and the grains are interconnected to form interconnected crystals, resulting in the formation of a dense structure inside the pellets, as shown in Figure 10c. The recrystallization process of the Fe_2_O_3_ crystal can be improved by roasting at the optimum temperature for enough time, and the fired pellet strength can be greatly improved. For hematite pellets, only when the temperature is high enough, do the Fe_2_O_3_ grains begin to recrystallize and grow up, forming a certain strength. However, such a recrystallization bond is not as strong as the recrystallization strength of Fe_3_O_4_ oxidized to Fe_2_O_3_ [105]. When the temperature rises above 1300 °C, the primary hematite will decompose and decrease in strength.

With the global decline in high-quality magnetite resources, pellet plants are forced to mix hematite with magnetite or use it alone to prepare pellet ores to meet the ever-increasing demand for steel smelting. However, both hematite pellets and pellets prepared with organic binders have obvious problems of insufficient fired strength. Many pellet researchers have tried to improve the solidification of mineral particles by adding a certain amount of veinstone components to promote the formation of a high-temperature liquid phase inside the pellet, thus, improving the fired pellet strength. Currently, inorganic mineral substances such as bentonite, calcined colemanite, boric acid, Na_2_CO_3_, NaCl, and copper slag are mainly added to pellet ores. There are different contents of SiO_2_, CaO, Al_2_O_3_, B_2_O_3_, Na^+^, and other chemical components in these substances, and the high-temperature heating process will react with each other to form a certain amount of low melting point liquid phase, which will enhance the contact between mineral particles, thus, significantly improving the pellet fired strength.

Due to its low price, excellent adsorption capacity, and swelling properties, bentonite has become the main additional component of inorganic pellet binder, which is applied to pellet ore in combination with various organic substances, such as starch, MHA, sodium lignosulfonate, and Alcotac^®^ CS. Not only can the addition of bentonite be effectively controlled within 1%, but it can also increase the strength of the fired pellet to 2500 N/pellet and above, which fully meets the basic strength requirements of pellet ores, as shown in Figure 11a–d. Bentonite mainly contains SiO_2_ and CaO, with SiO_2_ content up to approximately 70%. During the roasting process, SiO_2_ in bentonite does not react directly with Fe_2_O_3_ in hematite, but first reacts with CaO to form liquid-phase calcium silicate (CaO·SiO_2_), followed by a reaction with Fe_2_O_3_ to form liquid-phase calcium ferrate [108]. For magnetite pellets, SiO_2_ will react with CaO and Fe_3_O_4_ to form a certain amount of liquid-phase calcium silicate (CaO·SiO_2_) and iron olivine (2FeO·SiO_2_), respectively. The moderate addition of CaCO_3_, CaO, and other calcium-containing compounds helps to produce a high-temperature liquid phase, which is not only conducive to the diffusion of Fe^2+^ and Fe^3+^ and promotes Fe_2_O_3_ recrystallization, but also fills in the network structure formed by the interweaving of iron oxides and forms a slag phase connection, as shown in Figure 11e,f. The liquid phase enters the pores between mineral particles and reduces the porosity inside the pellet, enhancing the consolidation of the pellet and promoting a denser structure [109,110], as shown in Figure 11g,h. However, the excessive addition of bentonite will lead to excessive silicate production inside the pellet, increasing the viscosity of the liquid phase and reducing the fluidity [111], impeding Fe_2_O_3_ grain connection and inhibiting grain contact growth, thus, inhibiting the Fe_2_O_3_ recrystallization process and also leading to large area adhesion inside the pellet, shrinkage, and internal stress in the center of the pellet after cooling, forming many microcracks and leading to a decrease in the pellet strength [112]. Therefore, the addition of bentonite should be strictly limited during the pelletizing process to improve pellet strength and reduce metallurgical contamination.

In recent years, boron-containing compounds such as calcined colemanite, boric acid, and borax have been widely used as pellet additives to improve the strength of preheated and fired pellets [116,117], as shown in Figure 12a,b. Calcined colemanite and boric acid contain a large amount of B_2_O_3_, which has a low melting point and can generate a relatively large amount of liquid phase with CaO, SiO_2_, Al_2_O_3_, and other oxides at a lower temperature, which can not only reduce the liquid phase viscosity and pellet porosity, but also enhance the cementation between mineral particles, facilitate the diffusion of solid-phase mass points, promote Fe_2_O_3_ recrystallization, and significantly improve the fired pellet strength [118,119], as shown in Figure 12e–h. A moderate amount of B_2_O_3_ can form a more low melting point cementation phase, but a large amount of B_2_O_3_ will lead to an increase in the glass phases, and too much cementation phase will hinder the connection between the solid phase particles, the glass phase is brittle and has a low structural strength of its own, leading to a decrease in pellet strength [120].

The main sources of Na^+^ in pellets are Na_2_CO_3_, NaCl, and NaOH. Na_2_CO_3_ and NaOH may be introduced in the sodium process of calcium bentonite or residually mixed into pellets during the preparation of CMC, MHA, and other organic binders. Adding an appropriate amount of sodium salt to the pellet binder can effectively improve the fired pellet intensity, as shown in Figure 13a. Na^+^ can reduce the melting point of a high-temperature liquid phase, and it is easy to react with Fe_2_O_3_ to generate NaFeO_2_ and other low melting point liquid ferrites [125], which promotes a sufficient liquid phase to fill the pores of pellets, reduces the porosity of pellets, enhances the solid-phase consolidation, and improves the intensity of fired pellets [126], as shown in Figure 13c. However, too much Na^+^ will not only increase the amount of molten liquid phase at a high temperature, easily form agglomerated glass phase, and prevent the recrystallization process of Fe_2_O_3_, resulting in the reduction of pellet strength [127], but also leads to an increase in the lattice constant and cell volume of Fe_3_O_4_ during the reduction process, and thus an increase in the reduction swelling index(RSI) [128].

Cu-slag is a smelting tailing produced during the fire refining of copper using copper sulfide concentrates. For every 1 ton of copper concentrate produced, 2.0~2.2 tons of Cu-slag is generated, and the annual global emission of copper slag exceeds 20 million tons [130,131]. Cu-slag contains more Fe_3_O_4_, Al_2_O_3_, CaO, SiO_2_, and other oxides, which form a certain amount of CaAl_2_Si_2_O_8_ and Ca_3_Al_2_Si_3_O_12_ slag phases at high temperatures, enhancing the consolidation between iron ores, as shown in Figure 13d. FeO and Fe_3_O_4_ in Cu-slag enhance diffusion and recrystallization bonds during low-temperature heating, which helps to improve the denseness of pellet ore. After roasting at 1250 °C for 15 min, the strength of fired pellets with only 1.0wt% copper slag + 0.5wt% Na-lignosulfonate can reach 3100 N/pellet, as shown in Figure 13b. Cu-slag as a pellet additive not only improves pellet strength but also realizes the reuse of industrial waste resources, thus, saving resources and protecting the environment.

## 7. Conclusions

In recent years, after continuous research and exploration by pellet researchers, the types of organic binders have increased and the influence mechanism of organic polymers on pelletizing performance, bursting temperature, pellet strength at low and high temperatures, and other related theories have been constantly improved. An organic pellet binder can make pellets have excellent low-temperature performance and become the substitute for traditional bentonite due to its outstanding advantages of low metallurgical pollution. However, with the increase in temperature, the prominent problems of bursting and insufficient high-temperature strength of pellets prepared by organic binders have become the main factors that prevent the large-scale use of organic polymers in pellet production.

The main reason for the high-temperature burst of pellets prepared by organic binder is that the organic polymer makes the internal steam of pellets not diffuse to the surface in time, resulting in excessive internal steam pressure, which leads to uneven surface vaporization and internal diffusion of pellets. The insufficient high-temperature strength of pellets mainly includes the following two reasons: (1) organic polymers as hydrocarbons are not resistant to heat and are easily volatilized at high temperatures, leaving a large number of pores inside the pellets; (2) organic polymers do not contain inorganic components and hardly produce any liquid phase at high temperatures, resulting in the lack of solid phase consolidation between mineral particles. Currently, the solid-phase bridging between mineral particles is enhanced by adding small amounts of inorganic minerals, such as bentonite, borates, sodium salts, and copper slag, to produce a sufficient high-temperature liquid phase inside the pellet, which makes the fired pellet strength significantly improved. This treatment improves the high-temperature performance of the pellets, but at the same time, regenerates the slag phase in the pellet, leading to metallurgical contamination. However, this method can reduce the proportion of inorganic pellet binder represented by bentonite in the pellet to a certain extent and decrease metallurgical pollution and pellet additive cost. Therefore, composite pellet binders formed by compounding organic polymers and inorganic minerals are gradually becoming a new field of pellet binder research at present.

Nonetheless, although the composite pellet binder has achieved the performance of an inorganic pellet binder and an organic binder to complement each other, there are still some outstanding problems, such as the single synthesis method, large additive content, serious metallurgical pollution, and high cost. At present, under the premise of meeting the basic production requirements of pellets, the studies of most pellet researchers mainly focus on using valuable industrial solid waste to replace bentonite partially or completely as far as possible to reduce the amount of bentonite added and the cost of composite pellet additives and to protect the environment and promote the reuse of waste resources. At the same time, some pellet researchers try to prepare pellet ore by synthesizing geopolymers, but because of the imperfect research technology and theory, it still needs continuous innovative exploration and improvement.

## Figures and Tables

**Figure 1 polymers-14-04874-f001:**
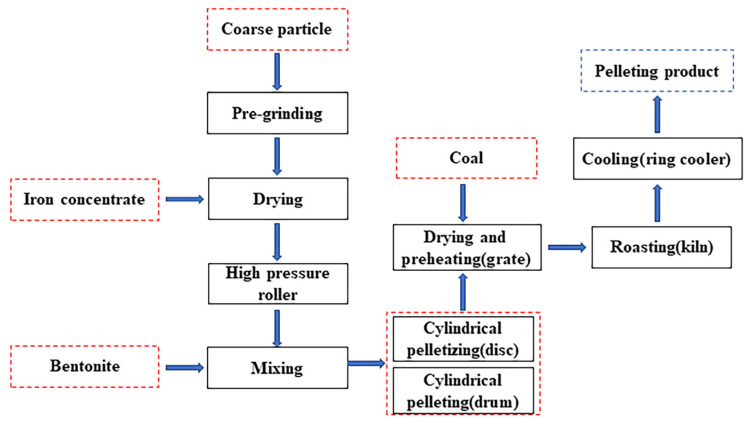
The process of producing iron ore pellets in factories of BAOWU [19].

**Figure 2 polymers-14-04874-f002:**
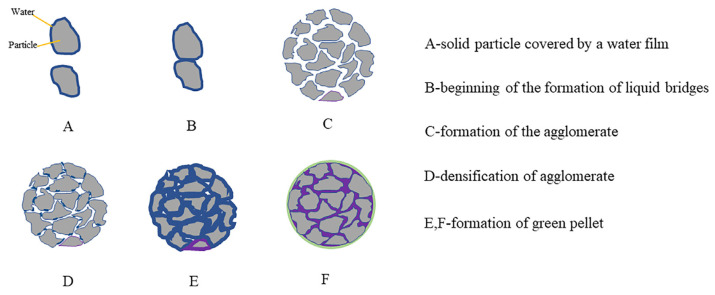
The steps in the formation of green pellets [20].

**Figure 3 polymers-14-04874-f003:**
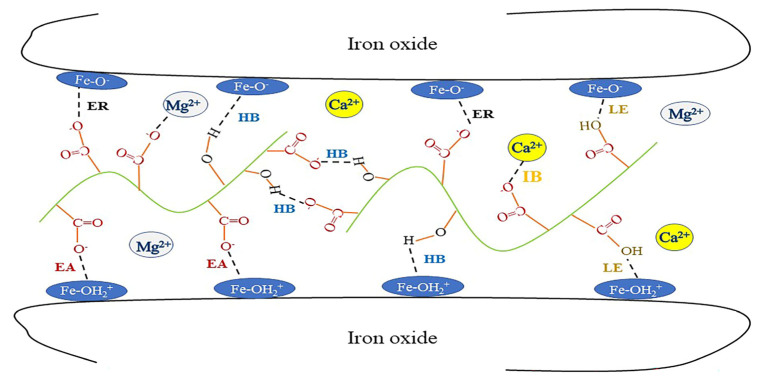
The schematic representation of surface interactions between iron oxide and binder. IB: ionic bond; HB: hydrogen bond; EA: electrostatic attraction; ER: electrostatic repulsion; LE: ligand exchange.

**Figure 4 polymers-14-04874-f004:**
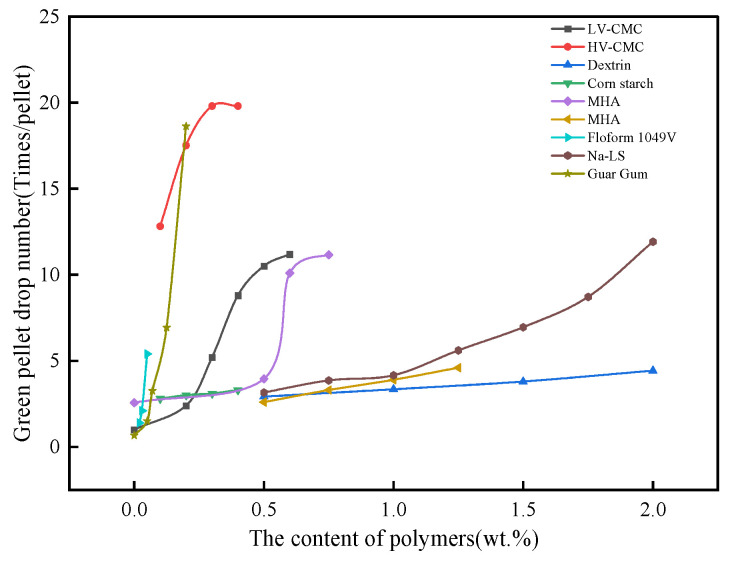
The effect of organic polymers on the green pellet drop strength [51,52,53,54,55,56,57,58,59].

**Figure 5 polymers-14-04874-f005:**
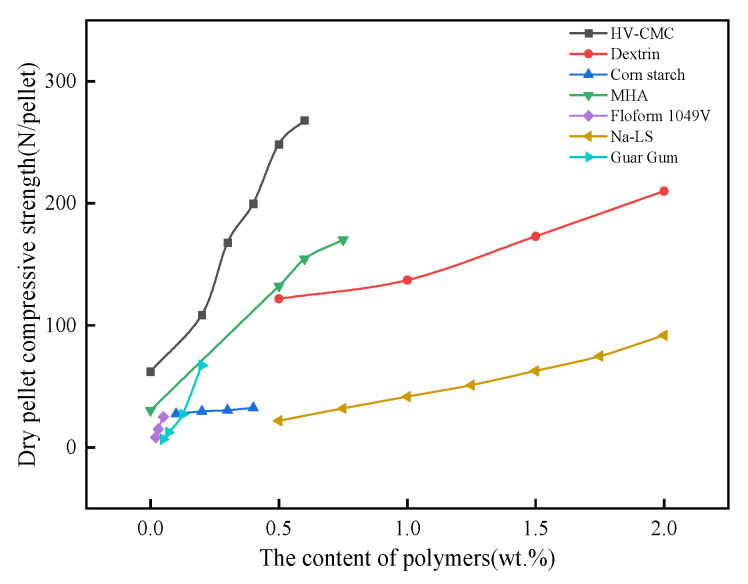
The effect of organic polymers on the dry pellet compressive strength [53,54,55,56,58,59,60].

**Figure 6 polymers-14-04874-f006:**
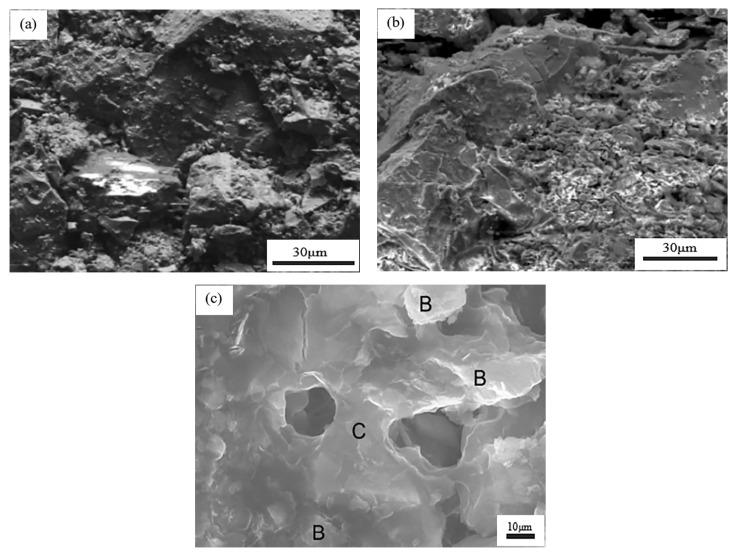
SEM image of the inner structure of a green pellet (**a**) without binder; (**b**) Gelatinized starch [73]; (**c**) Hematite pellets are made with 6.6 kg/ton starch. (B: bound mineral grain, C: starch gel matrix) [74].

**Figure 7 polymers-14-04874-f007:**
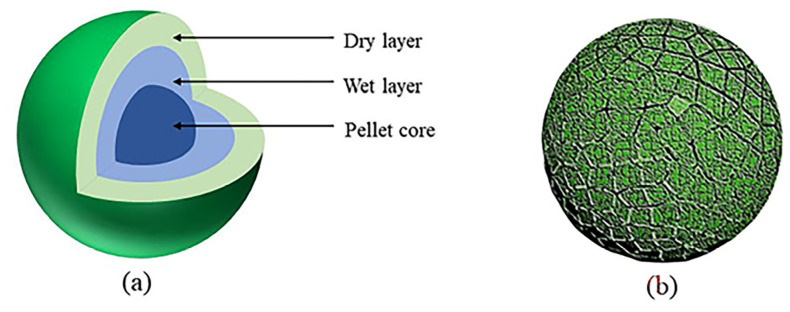
The 3D Model images of (**a**) a green pellet; (**b**) cracking pellet at high temperature.

**Figure 8 polymers-14-04874-f008:**
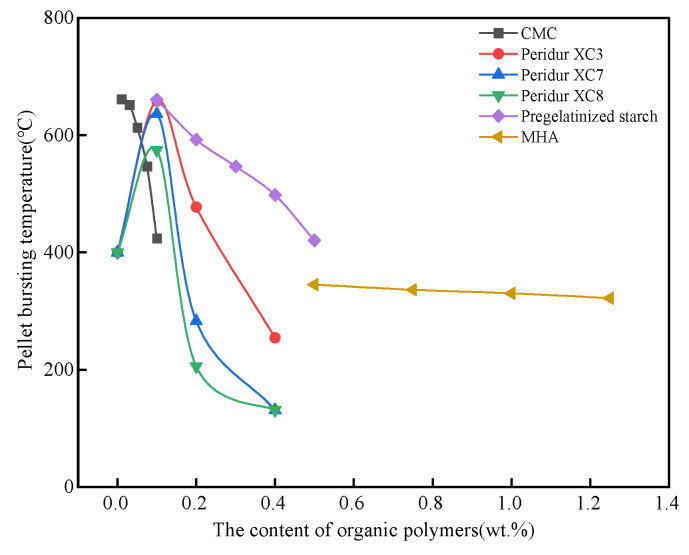
The effect of organic polymers on the pellet bursting temperature [96,97,98,99].

**Figure 9 polymers-14-04874-f009:**
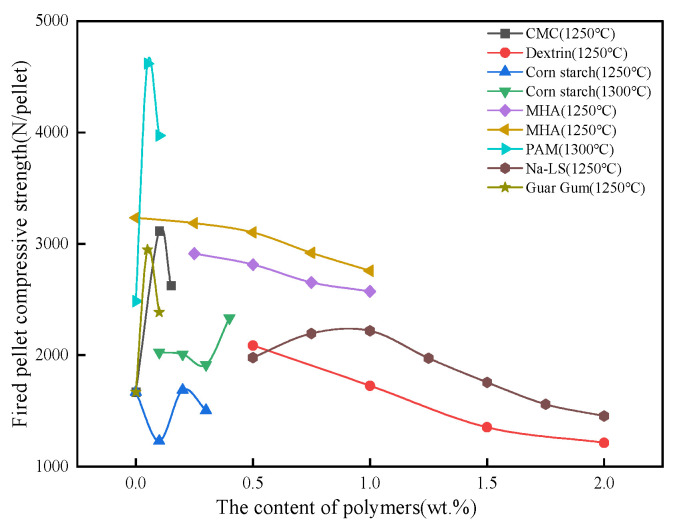
The effect of organic polymers on the fired pellet compressive strength [54,55,60,100,102,103,104].

**Figure 10 polymers-14-04874-f010:**
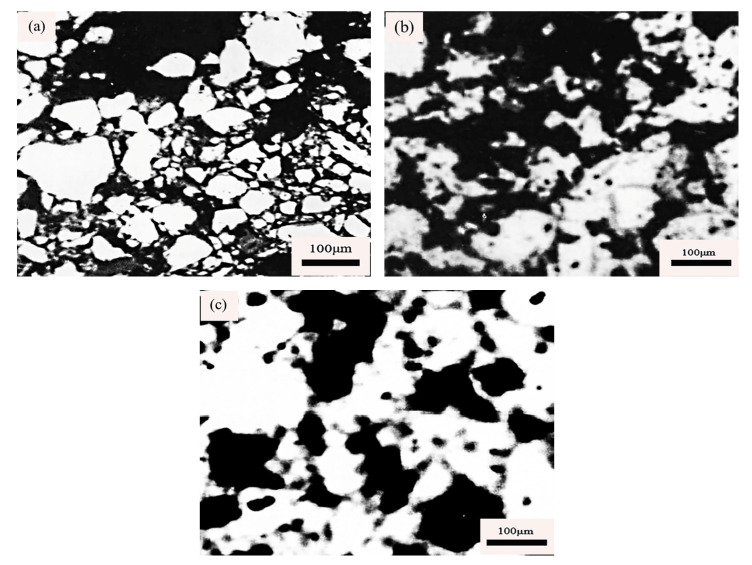
Internal grain microstructure of fired pellets: (**a**) individual primary crystals; (**b**) developing crystals; (**c**) interconnecting crystals [106,107].

**Figure 11 polymers-14-04874-f011:**
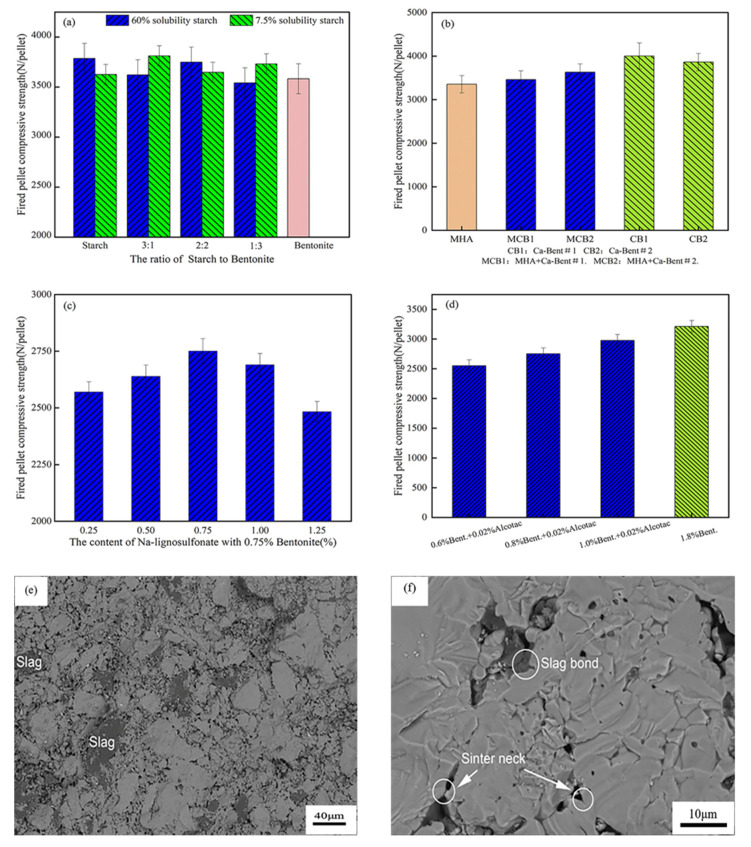
The effect of bentonite on the fired pellet compressive strength with (**a**) Kept the starch and bent. at 6.6 kg/t [113]. (**b**) The binder dose was 0.5wt%. MCB = 0.333wt% MHA + 0.167wt% Ca-bent. [52]. (**c**) Kept 0.75% bent. (**d**) Alcotac^®^ CS: Modified anionic polyacrylamide blend [114]. SEM images of fired pellets with (**e**,**f**) 2% bent. [115]. (**g**) 0.6wt% bent + 0.02wt% Alcotac. Porosity = 25%. (**h**) 1.8wt% bent. Porosity = 21% [114].

**Figure 12 polymers-14-04874-f012:**
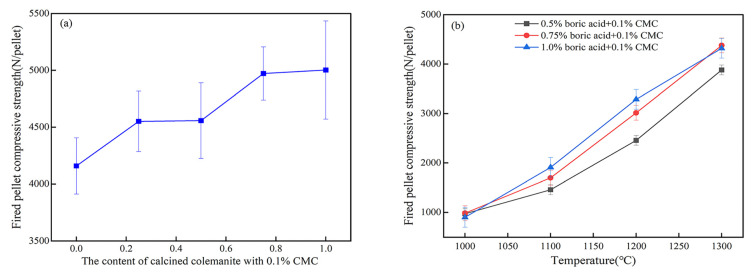
The effect of calcined colemanite or boric acid on the fired pellet compressive strength with (**a**–**d**) CMC and corn starch [55,121,122,123]. SEM images of fired pellets with (**e**) no binder. Porosity = 15.31%. (**f**) 0.4% Borax. Porosity = 11.52% [124]. (**g**) 0.50% calcined colemanite. (**h**) 0.10% corn starch + 0.50% calcined colemanite [55].

**Figure 13 polymers-14-04874-f013:**
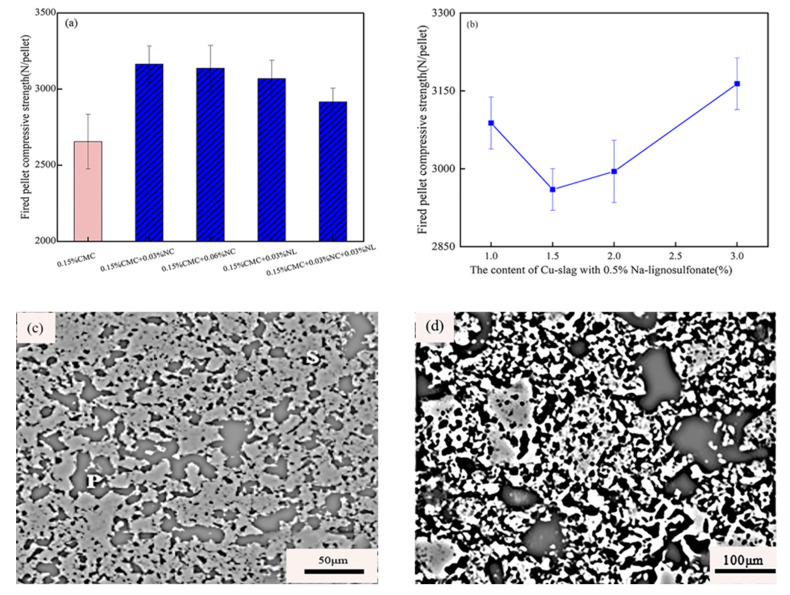
The effect of NaCl/Na_2_CO_3_ and Cu-slag on the fired pellet strength with (**a**) CMC (NL = NaCl, NC = Na_2_CO_3_) [102]. (**b**) Kept 0.5% Na-lignosulfonate [129]. SEM images of fired pellets with (**c**) 0.6% (Na_2_O + K_2_O) (P = Porosity, S = Slag) [126]. (**d**) 0.5%Na-lignosulfonate + 3%Cu-Slag [129].

## Data Availability

Not applicable.

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
