# Peer review of "A Review on the Effect of the Mechanism of Organic Polymers on Pellet Properties for Iron Ore Beneficiation"

_polymers, 2022, doi:10.3390/polym14224874_

Round 1
Reviewer 1 Report
The authors present a review about the effect mechanism of organic polymers on the iron ore pellet properties, from my point of view, there is minor somethings that should be improved to be published in this journal:
1. Perhaps the title is not appropriate, it can be: “A review of the effect mechanism of organic polymers on the Iron ore pellet properties”, I do not understand why use the term “special”
2. In section 4, in figures 4 and 5, it presents information on the effect of using various organic polymers and later makes a description of the characteristics of those polymers, but I think the order should be reversed.
3. In sections 5 and 6, in Figures 8 and 9 shows the effect of organic polymers on the pellet bursting temperature for some polymers but do not explain anything about the chemical composition of these polymers. I think it is necessary to go deeper into the discussion about the composition of polymers and their impact on the iron ore pellet properties, considering the focus of this journal.
Author Response
Response to Reviewer 1 Comments
The authors present a review of the effect mechanism of organic polymers on the iron ore pellet properties, from my point of view, there is minor somethings that should be improved to be published in this journal:
- Perhaps the title is not appropriate, it can be: “A review of the effect mechanism of organic polymers on the Iron ore pellet properties”, I do not understand why use the term “special”
Response 1: Thank you very much for receiving your suggestions. We have made comprehensive reference to your and other reviewers' comments on title modification. The title of the article will be changed to "A review on the effect of mechanism of organic polymer on pellet properties for iron ore beneficiation."Thank you very much for your professional level and for carefully reviewing our paper, sincerely thank you for your valuable suggestions.
- In section 4, in figures 4 and 5, it presents information on the effect of using various organic polymers and later makes a description of the characteristics of those polymers, but I think the order should be reversed.
Response 2: Thank you very much for receiving your suggestions. In Section 4, the mechanism of the effect of organic polymers on the low-temperature strength of pellet ore is mainly described. It demonstrates the effect of organic polymers on the strength of green pellets in Figure 4. It demonstrates the effect of organic polymers on the strength of dry pellets in Figure 5. We address the degree of influence of organic binder on pellet ore strength and then elaborate on the mechanism of influence of organic polymer inside the pellet ore. Reversing the order may be better for the reader to highlight the importance of the mechanism, but may overlook the extent to which the organic binder influences the pellet ore strength datum. Thank you very much for your professional level and for carefully reviewing our paper, sincerely thank you for your valuable suggestions.
- In sections 5 and 6, in Figures 8 and 9 show the effect of organic polymers on the pellet bursting temperature for some polymers but do not explain anything about the chemical composition of these polymers. I think it is necessary to go deeper into the discussion about the composition of polymers and their impact on the iron ore pellet properties, considering the focus of this journal.
Response 3: Thank you very much for receiving your suggestions.
It is well known that the vast majority of organic polymers are not resistant to high temperatures and the underlying reason is the lack of high-temperature-resistant groups and molecular backbones in the molecular structure. Figure 8 shows the effect of organic polymers on the bursting temperature of green pellets and Figure 9 shows the effect of organic polymers on the strength of fired pellets. Comparing Fig. 8 and Fig. 9, it is obvious that the green pellet bursting temperature and the fired pellet strength show the same trend of increasing and then decreasing with the addition of organic polymers.
In Section 5, we have reworked and reformatted the article with references to new literature. A brief description of the trend of the effect of organic matter on pellet bursting temperature in Figure 8 is presented, while the mechanism of pellet rupture generation is re-extended in the context of bentonite.
In Section 6 (Lines 475-489), The influence of polymers on the pelletizing properties of iron ore was further discussed in combination with the changing trend of the influence of organic polymer on the pelletizing strength. we elaborate on the main reasons for the lack of compressive strength of fired pellets prepared from organic polymers.
Under the absolute conditions of high temperatures, we concluded that it is of little theoretical significance to explain the phenomenon of deficient green pellet bursting temperature and fired pellet strength from the chemical structure of organic polymers.
Thank you very much for your professional level and for carefully reviewing our paper, sincerely thank you for your valuable suggestions.

Reviewer 2 Report
Dear Authors,
I find the article very interesting and surely other audience will too find it so. This review summarizes the use of various organic polymers in pellets.
It appears to me that the terms high temperature and low temperature are used in a loose way. Proposing some numerical boundaries would be commendable. Of course, a degree of arbitrariness will be hard to avoid entirely.
In my opinion, it should be stressed in the introduction that there are other comprehensive reviews published earlier—many of which are surely cited in the references of this manuscript. However, it would be welcomed by the readers a word or two about the differences and similarities to those other reviews. The introduction would be a good place. The comparison would give the reader a better perspective of the present review and only make your review more valuable.
Possibly the authors would want to refer to Fig. 5 when describing Fig. 7 and 9. In the latter two it is observed that the reduction in organic polymer content percentage increases both, pellet bursting temperature and pellet compressive strength--mostly monotonically, save for Na-LS. And there is another group: CMC, guar gum and PAM, have a raising and then dropping profile--as a function of organic polymer reduction. Then a third group for starch, which drops and rises or zigzags. It would be interesting to have some discussion about those trends. Here I refer to Fig. 9 but similar observations apply to Fig. 7
It is unfortunate that the materials included in both figures 7 and 9, are not the same. If they were the comparison would be more complete.
The phenomenological description of bursting states succinctly the main elements at play. Compositional details will of course be decisive in the temperature at which bursting happens. Figure 7 gives a graphical description of trends. As commented above, probably the authors should exploit a bit more those trends and extend the discussion.
In the conclusions the authors mentioned the use of geopolymer to prepare pellet ore, but geopolymers were never mentioned earlier in the text. That aspect should be also discussed earlier in the text for completeness' sake.
Lastly, although cosmetic, I would advise to be more consistent about punctuation. Often times blanks are missing and other punctuation elements.
Best of luck with the publication process.
Author Response
Response to Reviewer 2 Comments
Dear Authors,
I find the article very interesting and surely other audience will too find it so. This review summarizes the use of various organic polymers in pellets.
It appears to me that the terms high temperature and low temperature are used in a loose way. Proposing some numerical boundaries would be commendable. Of course, a degree of arbitrariness will be hard to avoid entirely.
- In my opinion, it should be stressed in the introduction that there are other comprehensive reviews published earlier—many of which are surely cited in the references of this manuscript. However, it would be welcomed by the readers a word or two about the differences and similarities to those other reviews? The introduction would be a good place. The comparison would give the reader a better perspective of the present review and only make your review more valuable.
Response 1: Thank you very much for receiving your suggestions. We followed your suggestion and added the difference between this review and other related review articles in the third paragraph of the introduction section (Lines 98-106). Thank you very much for your professional level and for carefully reviewing our paper, sincerely thank you for your valuable suggestions.
- Possibly the authors would want to refer to Fig. 5 when describing Fig. 7 and 9. In the latter two, it is observed that the reduction in organic polymer content percentage increases both, pellet bursting temperature and pellet compressive strength--mostly monotonically, save for Na-LS. And there is another group: CMC, guar gum and PAM, have a raising and then dropping profile--as a function of organic polymer reduction. Then the third group for starch, which drops and rises or zigzags. It would be interesting to have some discussion about those trends. Here I refer to Fig. 9 but similar observations apply to Fig. 7. It is unfortunate that the materials included in both figures 7 and 9 are not the same. If they were the comparison would be more complete.
Response 2: Thank you very much for receiving your suggestions. In Figs. 8 and 9, for the effect of organic polymers on pellet bursting temperature and fired pellet strength, we discuss in more depth the mechanism of polymer effects on iron ore pellet properties about the trends of the different substances in the plots, as are shown in Lines 361-462 and Lines 473-489. Thank you very much for your professional level and for carefully reviewing our paper, sincerely thank you for your valuable suggestions
- The phenomenological description of bursting states succinctly the main elements at play. Compositional details will of course be decisive in the temperature at which bursting happens. Figure 7 gives a graphical description of trends. As commented above, probably the authors should exploit a bit more those trends and extend the discussion.
Response 3: Thank you very much for receiving your suggestions. In chapter 5, we revised a lot of content and typesetting order. The influence trend of organic matter on pellet bursting temperature in Fig. 8 was briefly described, and the mechanism of pellet bursting was extended with bentonite(Lines 361-462). Thank you very much for your professional level and for carefully reviewing our paper, sincerely thank you for your valuable suggestions.
- In the conclusions, the authors mentioned the use of geopolymer to prepare pellet ore, but geopolymers were never mentioned earlier in the text. That aspect should be also discussed earlier in the text for completeness' sake.
Response 5: Thank you very much for receiving your suggestions. In recent years, geopolymer is a new technology for pellet ore preparation, which also has many drawbacks at present, but can significantly reduce the percentage of pellet binder addition and improve the strength of pellet ore. The reference to geopolymers at the conclusion is an outlook on the latest trends in existing metallurgical technology, but it is not part of the organic pellet binder. We hope that the reviewers will allow us to keep the elaboration of geopolymers in the conclusions. Thank you very much for your professional level and for carefully reviewing our paper, sincerely thank you for your valuable suggestions.
- Lastly, although cosmetic, I would advise being more consistent about punctuation. Often times blanks are missing and other punctuation elements are. Best of luck with the publication process.
Response 6: Thank you very much for receiving your suggestions. We have proofread and corrected similar errors in the article as you suggested. Thank you very much for your professional level and for carefully reviewing our paper, sincerely thank you for your valuable suggestions.

Reviewer 3 Report
The paper is well written, but only a few comments to be addressed. It is also not my duty to correct the grammar. The paper must be thoroughly read to correct all grammatical errors and typos. The paper should be accepted if the authors are able to address the comments annotated in the PDF file. Please also check my suggestion for the modification of the topic.
1.The authors kept saying Green pellets. What do they mean by green pellets?
2. What id the difference between this green and dry pellets?
3. Can the authors come out clearly the Limitations or the advantages and the disadvantage of using the already existing bentonite and the organic polymers as a replacement to the clay binder.

Author Response
Response to Reviewer 3 Comments
The paper is well written, but only a few comments to be addressed. It is also not my duty to correct the grammar. The paper must be thoroughly read to correct all grammatical errors and typos. The paper should be accepted if the authors are able to address the comments annotated in the PDF file. Please also check my suggestion for the modification of the topic.
- The authors kept saying Green pellets. What do they mean by green pellets?
Response 1: Thank you very much for receiving your questions. "Green pellet" is a term used in the discipline of metallurgical engineering. "Green pellet" refers to a pellet of a certain diameter, strength, and moisture content formed by the pellet-making process of a disc pelletizing machine. The production of pellet ore passes through four stages: green pellet, dry pellet, preheated pellet, and fired pellet, while the green pellet is the most important basic production stage. Thank you very much for your professional level and for carefully reviewing our paper, sincerely thank you for your valuable questions.
- What is the difference between these green and dry pellets?
Response 2: Thank you very much for receiving your questions. After mixing the iron ore powder and pellet additives thoroughly, the pellet binder absorbs water to bond the iron ore powder under the continuous rotation of the disc pelletizing machine, and gradually forms green pellets of a certain diameter size. To facilitate the transfer, the low-strength green pellets need to be baked at 105°C for a certain period to reach the pellet strength requirement and facilitate further preheating and roasting of the dry pellets in the shaft furnace. Thank you very much for your professional level and for carefully reviewing our paper, sincerely thank you for your valuable questions.
- Can the authors come out clearly the Limitations or the advantages and the disadvantage of using the already existing bentonite and the organic polymers as a replacement to the clay binder?
Response 3: Thank you very much for receiving your questions. On these points, we have the following ideas:
Bentonite is the traditional pellet binder and is still commonly used in many small, medium, and large pellet plants. Bentonite has the advantages of relatively low price, high water absorption, strong bonding performance, and high strength of fired pellets, but bentonite also has the following shortcomings.
- add a large proportion. As high-quality sodium bentonite resources continue to decline, its price is also increasing, so some pellet plants are forced to use poor-quality calcium bentonite for economic benefits, which never led to a large proportion of bentonite added to the problem, some pellet plants in China add a proportion of up to 6 wt.%, much higher than the proportion of foreign additions.
- serious metallurgical pollution. Bentonite after high-temperature sintering will produce a large number of inorganic silicate phases inside the pellet ore. A large amount of steel slag will be produced in the reduction process, which increases the energy input.
Compared with bentonite, an organic polymer as a new type of pellet binder has the advantages of high water absorption, strong bonding performance, and fast pelletizing rate, but also has the following disadvantages.
- High cost. Most of the organic polymers are chemically modified to have certain adsorption and bonding properties, so the cost has been higher, which also limits its addition to the pellet.
- Low strength of fired pellets. Most of the organic polymers are hydrocarbons, which are volatilized by thermal decomposition in the high-temperature sintering stage without producing any silicate slag phase inside the pellet ore, lacking solid-phase bridging effect, while the porosity inside the pellet will increase with the addition of organic binder, thus leading to insufficient strength of fired pellets.
We hope the descriptions can answer your deep concerns and help our readers to understand the advantages and the disadvantage of using the already existing bentonite and organic polymers. Thank you very much for your professional level and for carefully reviewing our paper, sincerely thank you for your valuable questions.
